# A Pediatric Case of Inverted Meckel’s Diverticulum Presenting with Cyclic Vomiting-like Symptoms: A Case Report and Literature Review

**DOI:** 10.3390/children9121817

**Published:** 2022-11-24

**Authors:** Yoshiko Endo, Keisuke Jimbo, Nobuyasu Arai, Takanori Ochi, Mitsuyoshi Suzuki, Atsuyuki Yamataka, Toshiaki Shimizu

**Affiliations:** 1Department of Pediatrics, Juntendo University Faculty of Medicine, Tokyo 113-8421, Japan; 2Department of Pediatric General and Urogenital Surgery, Juntendo University Faculty of Medicine, Tokyo 113-8421, Japan

**Keywords:** abdominal ultrasonography, ileo-ileal intussusception, double balloon endoscopy, intestinal tumor, inverted Meckel’s diverticulum

## Abstract

Asymptomatic Meckel’s diverticulum cases are not uncommon, leading to diagnostic difficulties in cases of atypical presentations with only gastrointestinal symptoms other than bloody stool. A nine-year-old boy diagnosed as having cyclic vomiting because of recurrent abdominal pain and vomiting for 6 months was referred to our institute and hospitalized due to worsening symptoms. After admission, abdominal ultrasonography showed the multiple concentric ring sign and a pseudokidney sign at the lower ileum, leading to the diagnosis of ileo-ileal intussusception, but the gastrointestinal symptoms and ultrasonic findings disappeared spontaneously. Transanal, double-balloon, intestinal endoscopy demonstrated a pedunculated polyp-like structure, and surgical resection was performed. An inverted diverticulum was found in the resected intestinal lumen, and ectopic gastric mucosa was identified histologically, leading to the diagnosis of inverted Meckel’s diverticulum. In pediatric cases involving periodic attacks of vomiting and abdominal pain, unnecessary emergent surgery could be avoided by cautious imaging evaluation and consideration of ileo-ileal intussusception with advanced lesions of an inverted Meckel’s diverticulum as a differential diagnosis, without facilely diagnosing cyclic vomiting. In addition, previous reports of inverted Meckel’s diverticulum were reviewed, and the results were compared between adult and pediatric groups in each category.

## 1. Introduction

Typical cases of Meckel’s diverticulum are usually diagnosed by gastrointestinal symptoms such as acute abdominal pain, vomiting, and, especially, bloody stool. However, asymptomatic Meckel’s diverticulum cases are not uncommon, since approximately only 30% of Meckel’s diverticulum cases show clinical symptoms throughout their life, leading to diagnostic difficulties in cases of atypical presentations with only gastrointestinal symptoms other than bloody stool [1]. Therefore, there are only a few cases of children with abdominal pain and vomiting without bloody stool not undergoing emergency surgery for ileo-ileal intussusception, despite a relatively long history from disease onset [2]. Meckel’s diverticulum was also found to be a risk factor for ileo-ileal intussusception in children and adults due to occasional inversion [2,3]. On the other hand, in recent years, a few reports have demonstrated the efficacy of double-balloon endoscopy (DBE) for the diagnosis of hemorrhage in pediatric Meckel’s diverticulum [4,5]. Although transanal DBE was considered suitable for children over 3 years of age and weighing at least 14 kg [6], no reports of an inverted Meckel’s diverticulum identified by DBE prior to surgery without gastrointestinal bleeding have been identified to date.

In this report, a case of cyclic vomiting-like symptoms such as periodic abdominal pain and vomiting with inverted Meckel’s diverticulum diagnosed using transanal DBE after abdominal ultrasonography is presented. Previous reports of inverted Meckel’s diverticulum were reviewed, statistics such as median values were calculated, and comparisons between adults and children were performed according to each parameter.

## 2. Case Report

### 2.1. Patient Information

The patient was a 9-year-old boy with no relevant past or family history.

### 2.2. Symptoms and Clinical Findings

The patient was admitted to our institute with the chief complaint of a six-month history of recurrent abdominal pain and vomiting. He had been diagnosed as having cyclic vomiting by his family physician and referred to our institute for further treatment. At the time of referral, the patient was in the intermittent phase, and the fecal hemoglobin (Hb) level was 29 [reference range (RR) 0–100] ng/mL. Therefore, considering the possibility that the cyclic vomiting may resolve during the natural course of the disorder, the patient was followed without treatment. However, his symptoms worsened two days before admission, and he was admitted to our institute on the same day he visited as an outpatient.

At admission, he was 133.3 cm (74.8th percentile) tall and weighed 28.3 kg (50.1st percentile), with weight loss of 2.7 kg over three months. However, no reduction of height gain was observed on the growth curve. The patient’s axillary temperature was 36.0 °C; his pulse was 60 beats/min; and his blood pressure was 93/49 mmHg. On physical examination, his facial color was slightly pale, but no obvious abnormalities of the eyelid conjunctiva, oral cavity, chest, or skin were found. His abdomen had normal bowel peristalsis and no distention, mass, or guarding, but there was mild tenderness of the lower mid-abdomen.

Laboratory findings at admission were as follows: hemoglobin 13.9 (RR 12.6–16.5) g/dL and ferritin of 46 (RR 25–280) ng/mL, indicating no iron deficiency anemia; total protein 6.5 (RR 6.3–7.8) g/dL and albumin 4.4 (RR 3.8–4.8) g/dL within normal ranges; and inflammatory markers not increased. On the other hand, only fecal human Hb was mildly elevated (232 ng/mL).

On imaging, the plain abdominal X-ray showed no remarkable abnormalities, but abdominal ultrasonography showed the multiple concentric ring sign and the pseudokidney sign in the ileum, which were not released spontaneously by intestinal peristalsis. A mass-like structure was observed at the tip of the stacked inner tract, and Doppler imaging showed a marked blood signal concentrating toward the tip. Based on the above findings, ileo-ileal intussusception with some pathological advanced lesions was diagnosed (Figure 1).

After admission, the patient was kept fasted, and fluid replacement was started. After approximately two hours, the abdominal pain and vomiting resolved spontaneously with the disappearance of the pseudo-kidney sign on abdominal ultrasonography. On the second day of admission, DBE was performed under sedation, considering that normal colonoscopy could not reach the lesion based on the ultrasonographic findings, and a stalked polyp-like structure was detected at approximately 90 cm proximal to the ileocecal valve (Figure 2a,b). An inverted Meckel’s diverticulum was strongly suspected because of the atypical structure of the pedunculated polyp. Since endoscopic resection appeared to be extremely risky because of marked Doppler signal accumulation on abdominal ultrasonography, only position marking was performed to make it easy to find during operation.

### 2.3. Therapeutic Intervention

It was decided to perform semi-emergent surgery rather than emergent surgery because the patient had a temporary small intestinal intussusception with spontaneous resolution of symptoms on ultrasonographic and clinical findings, with similar episodes, and parental consent could not be obtained immediately. Approximately two weeks after the last vomiting episode, because the patient suffered no symptoms at all during this period, the patient underwent laparoscopic-assisted surgery by pediatric surgeons and was found to have a diverticulum on the anti-mesenteric side of the marked ileum that inverted into the intestinal tract at approximately 90 cm proximal to the ileocecal valve (Figure 2c). No bowel ischemia, necrosis, or abnormal amounts of ascites were observed. The area around the marked portion was incised, and a mass was found at the tip of the inverted tubular structure. After combined resection of the diverticulum and surrounding ileum, the ileum was anastomosed end-to-end extra-corporeally. An attempt was made to manipulate the inverted diverticulum immediately after bowel resection, but it could not be completed. Surgical resection was successfully performed without recurrence of gastrointestinal symptoms with oral feeding.

### 2.4. Follow-up and Outcome

The patient was diagnosed as having an inverted Meckel’s diverticulum based on the histological presence of ectopic gastric mucosa in the mass-like structure of the diverticulum (Figure 2d). Postoperatively, the recurrent gastrointestinal symptoms resolved, and they have not relapsed for more than four years.

## 3. Discussion

### 3.1. Literature Review

In order to understand inverted Meckel’s diverticulum, the PubMed database was searched from its inception to February 2022 using the keyword “inverted Meckel’s diverticulum”, and the search language was limited to English. In total, 94 related articles were retrieved. There were 29 articles for 74 adult [3,7,8,9,10,11,12,13,14,15,16,17,18,19,20,21,22,23,24,25] and 9 pediatric (age < 18 years) [2,26,27,28,29,30,31,32] cases including the present case] that reported the course of each case in detail. These included 55 male and 28 female patients with onset ages of 2 to 78 years. Clinical symptoms, duration from onset to treatment, positive diagnostic modalities, treatment, and pathological findings were compared between children and adults. Quantitative data were analyzed by Student’s *t*-test and the Mann–Whitney U test. For the categorical data, the χ^2^ test was performed when there was a sufficient number of cases, and Fisher’s exact test was performed for the analysis involving a relatively small number of columns. The significance level was *p* < 0.05 in the two-sided test. In the comparison between adult and pediatric cases, pediatric cases with an inverted Meckel’s diverticulum were significantly less likely to show melena (*p* = 0.019) and had clinical features tending to cause intussusception (*p* = 0.001). In addition, the pediatric cases were operated on within a significantly shorter time from onset than the adult cases (*p* = 0.019) (Table 1).

### 3.2. Short Summary

A pediatric case of inverted Meckel’s diverticulum diagnosed by abdominal ultrasonography and DBE, in which the patient was diagnosed as having cyclic vomiting due to recurrent abdominal pain and vomiting by a referring physician, was described. Initially, because of high fecal human Hb levels during abdominal pain and vomiting attacks, abdominal ultrasound was performed, and incomplete ileal intussusception was diagnosed with findings of multiple concentric ring signs and the pseudokidney sign, which resolved spontaneously during the disease course. Finally, laparoscopic-assisted resection of a pedunculated polyp-like structure identified at approximately 90 cm proximal to the ileocecal valve, which is a preferred site for Meckel’s diverticulum, led to the definitive diagnosis of inverted Meckel’s diverticulum with histological findings of ectopic gastric mucosa. After resection of the lesion, no recurrence of gastrointestinal symptoms has been observed.

### 3.3. Meckel’s Diverticulum

Meckel’s diverticulum is a residual structure of the embryonic yolk duct, and most cases are diagnosed by bloody stool and obstructive bowel symptoms such as abdominal pain and vomiting. Symptomatic Meckel’s diverticulum cases are estimated to account for about 30% of all Meckel’s diverticulum cases [1,33,34]. About 50% of symptomatic cases requiring a surgical procedure were reportedly diagnosed at less than five years of age, with sex ratios (M:F) between 1.5:1 and 4:1 [33]. In symptomatic or hemorrhagic cases of Meckel’s diverticulum, approximately 53% showed ectopic tissue, and approximately 80% of those were gastric mucosal tissue [35]. Therefore, the diagnosis may be difficult in cases with an atypical onset age and the only obvious clinical manifestation is abdominal pain, as in the present case. Indeed, the diagnosis of the present case was complicated by the fact that the fecal occult blood test was positive during the gastrointestinal attack, but no obvious bloody stool was evident, despite the ectopic gastric mucosal tissue that was present within the advanced mass lesion on histological examination. In the literature review including the present case, bloody stool was observed in 55/74 (74.3%) of adults and 3/9 (33.3%) of children, with a significant difference (*p* = 0.019) (Table 1). Although it may not be sufficient to conclude that the diagnosis of inverted Meckel’s diverticulum is usually delayed in pediatric cases because of the absence of bloody stool, eight pediatric cases, excluding the present case, resulted in ileus due to intussusception, and imaging findings of intussusception were observed significantly more often in children than in adults (*p* = 0.001) [2,3,7,8,9,10,11,12,13,14,15,16,17,18,19,20,21,22,23,24,25,26,27,28,29,30,31,32]. On the other hand, adult cases may have significantly more anemia due to chronic intestinal hemorrhage (*p* < 0.0001) (Table 1), because the adult cases do not present with the acute onset of intussusception, unlike the pediatric cases.

### 3.4. Clinical Course

The present patient had periodic abdominal pain and vomiting from the age of eight years, suggesting that the Meckel’s diverticulum inverted during the same period, and there was repeated ileo-ileal intussusception. Previous reports suggested that the symptoms of small intestinal intussusception were nonspecific and usually detected incidentally by abdominal CT or ultrasonography, with a high rate of spontaneous resolution. However, symptomatic cases were often difficult to treat noninvasively, leading to surgical intervention [36,37]. The present review also showed that 5/9 (62.5%) of the pediatric patients underwent emergency surgery within several hours to days after the onset of symptoms, and the median time from onset to surgery was significantly (*p* = 0.019) shorter in children [0.1 (0.02–6) months] than in adults [3.5 (0.03–60) months] (Table 1). Thus, in children, although the existence of a mechanism leading rapidly from inversion to irreversible intussusception was suggested, the elective procedure had been performed prior to the occurrence of irreversible intussusception in the present case.

### 3.5. Diagnosis and Treatment

A reason for the rapid diagnosis of the present case was that the positive fecal occult blood examination led to abdominal screening by ultrasonography. In the review, abdominal CT and ultrasonography were used in 42/74 (56.8%) and 2/9 (22.2%), respectively, of the adult patients and 14/74 (18.9%) and 8/9 (88.9%), respectively, of the pediatric patients. Abdominal ultrasonography was primarily used as a diagnostic modality in children, with a positive finding rate of 8/8 (100%) (Table 1). Therefore, the results suggest that abdominal ultrasonography is not only radiation-free and noninvasive compared to abdominal CT, but it is also a useful tool in the diagnosis of intussusception caused by advanced pathological lesions, including inverted Meckel’s diverticulum in children. On the other hand, a Meckel’s scan was performed in six adult patients including cases with ectopic gastric mucosa, but all cases had negative findings, indicating that a Meckel’s scan may not be effective in diagnosing inverted Meckel’s diverticulum. DBE was performed in six adult patients, and positive findings were observed in 6/6 (100%), but the present study was the first in which DBE was performed in a pediatric case. It was thus demonstrated that an inverted Meckel’s diverticulum could be clearly visualized by DBE even in children. However, endoscopic therapeutic resection of an inverted Meckel’s diverticulum in pediatric cases may not yet be practical because of the absence of endoscopic procedure case reports in children and the few institutions providing prompt surgical support in emergencies. However, the differential diagnosis of inverted Meckel’s diverticulum with a mass-like leading point, as in the present case, includes isolated small intestinal pedunculated polyps. In the case of small intestinal polyps, small intestinal balloon endoscopy is useful in performing polypectomy without invasive surgery [38]. In addition, the presence of small intestinal polyps may suggest the possibility of juvenile polyposis syndrome [38], and small intestinal balloon endoscopy may also contribute to screening for such a disorder.

## 4. Conclusions

A case of inverted Meckel’s diverticulum that mimicked cyclic vomiting due to recurrent abdominal pain and vomiting was presented. This case was promptly diagnosed by DBE, resulting in prevention of intestinal perforation and obstruction and exclusion of a small intestinal polyp and polyposis. In cases involving periodic attacks of vomiting and abdominal pain, unnecessary emergent surgery could be avoided by cautious imaging evaluation, considering ileo-ileal intussusception with an advanced lesion of an inverted Meckel’s diverticulum as a differential diagnosis, without facilely diagnosing cyclic vomiting. In addition, the novelty of this case consists of the fact that the inverted Meckel’s diverticulum did not result in an evolving intussusception in a child patient. Thus, the necessity of emergent surgery is not always related to a thorough clinical and imaging evaluation, but to the possibility of an inverted Meckel’s diverticulum causing an evolving intussusception, especially in adults. Thus, an inverted Meckel’s diverticulum alone is not an indication for emergency surgery, but complications of Meckel’s diverticulum such as unresolved intestinal intussusception with intestinal ischemia may prompt an emergent surgical exploration.

## Figures and Tables

**Figure 1 children-09-01817-f001:**
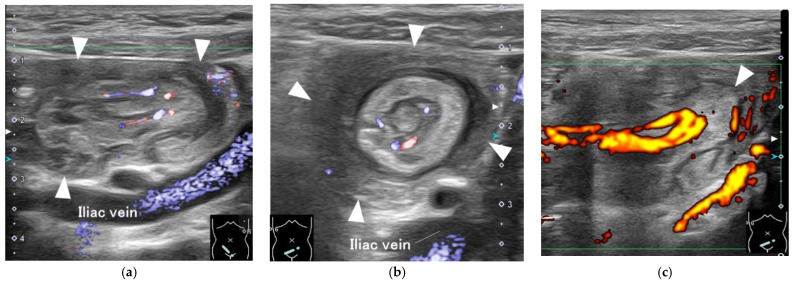
Ultrasonography findings at admission. (**a**) Pseudokidney sign in the lower abdominal long axis image (advanced dynamic flow); (**b**) Multiple concentric ring signs in the lower abdominal short axis image (advanced dynamic flow); (**c**) Increased blood flow signal towards the mass lesions (arrowhead) at the tip of the inner tube (power Doppler image).

**Figure 2 children-09-01817-f002:**
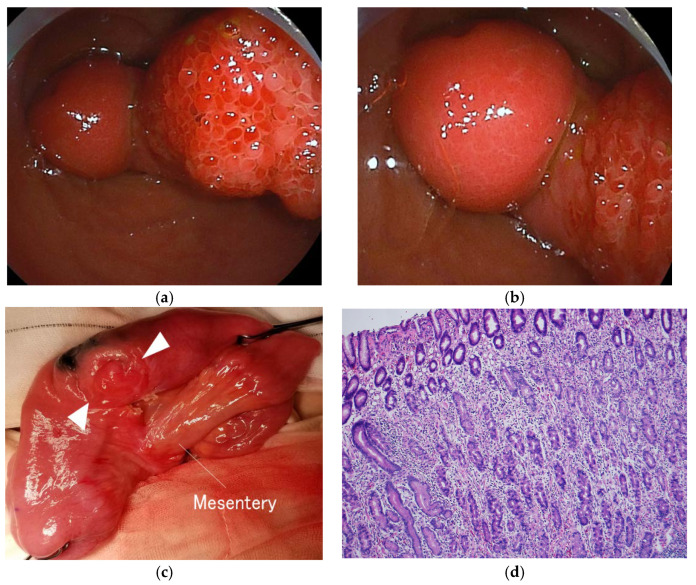
The endoscopic and histological findings of the inverted Meckel’s diverticulum. (**a**) Double-balloon endoscopy shows a pedunculated polyp-like structure; (**b**) A mass lesion is found at the tip of the polyp-like structure; (**c**) An inverted diverticulum (arrowhead) is found on the opposite side of the mesenteric attachment of the resected section; (**d**) Ectopic gastric mucosa is found on histological examination of the mass lesion at the tip (HE stain, ×100).

**Table 1 children-09-01817-t001:** Comparison of adult and pediatric cases of inverted Meckel’s diverticulum in the literature review.

Reviewed Items	Adult	Pediatric	*p* Value
Age (y) (range)	42 (18–78)	8 (2–16)	<0.001
Female (%)	24/74 (32.4)	4/9 (44.4)	0.48
Clinical signs and symptoms (%)			
Abdominal pain/vomiting	48/74 (64.9)	9/9 (100)	0.051
Bloody stool	55/74 (74.3)	3/9 (33.3)	0.019
Anemia	56/74 (75.7)	0/9 (0)	<0.0001
Intussusception	31/74 (41.9)	9/9 (100)	0.001
Time from onset to treatment (months) (range)	3.5 (0.03–60)	0.1 (0.02–6)	0.019
Positive diagnostic modality (%)			
Abdominal CT scan	42/42 (100)	2/2 (100)	>0.99
Meckel’s scan	0/6 (0)	0/0 (0)	>0.99
Abdominal ultrasonography	12/14 (85.7)	8/8 (100)	0.52
Small intestinal capsule endoscopy	3/4 (75.0)	1/1 (100)	>0.99
Double-balloon endoscopy	6/6 (100)	1/1 (100)	>0.99
Treatments (%)			
Diverticulectomy/partial ileal resection	72/74 (97.3)	9/9 (100)	>0.99
Endoscopic resection with DBE	2/74(2.7)	0/9 (0)	>0.99
Pathological finding (%)			
Ulceration	47/74 (63.5)	3/9 (33.3)	0.15
Ectopic gastric mucosa	30/74 (40.5)	2/9 (22.2)	0.47
Ectopic pancreas	18/74 (24.3)	5/9 (55.6)	0.11

## Data Availability

The data presented in this study are available on request from the corresponding author. The data are not publicly available due to privacy.

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
