# Peer review of "A Pediatric Case of Inverted Meckel’s Diverticulum Presenting with Cyclic Vomiting-like Symptoms: A Case Report and Literature Review"

_children, 2022, doi:10.3390/children9121817_

Round 1

Reviewer 1 Report

In this age group, patients with intussusception usually have leading points. In your case, it is inverted meckel's diverticulum. There are possibilities of bowel ischemia or even perforation if left untreated. There are several questions I want to ask you. 

1. Is there any signs of bowel ischemia or necrosis when you do the operation? Are there any ascites?

2. Is the operation done by total laparoscope? Including intra-coporeal end to end anastomosis? Is there any discrepancy between the proximal and distal bowel? 

3. Why did you choose to do the operation about two weeks later? Why didn't you do the operation urgently? Do you have to worry about the possibility of bowel ischemia or worsening of the patient's condition?

4. Because a pathologic leading point is highly suspected in this case, do you think double-balloon endoscopy is necessary? Is the exam will affect your further treatment plan? If the DBE is negative in this case, what will you do next? Is there any suggestion you want to give to the readers?

Author Response

Reviewer 1

In this age group, patients with intussusception usually have leading points. In your case, it is inverted Meckel's diverticulum. There are possibilities of bowel ischemia or even perforation if left untreated. There are several questions I want to ask you. 

Response: Thank you very much for your helpful comments on our paper. We have revised our manuscript based on the suggestions provided. The corrections are shown in red in this letter and in our revised manuscript. We hope that you will find these revisions satisfactory.

Specific Comments:

  1. Is there any signs of bowel ischemia or necrosis when you do the operation? Are there any ascites?

Response: No bowel ischemia or necrosis was found at the time of operation, and no unusual amounts of ascites were observed. We have revised our manuscript to include the surgical findings as follows:

Page 3, lines 111-112: No bowel ischemia, necrosis, or abnormal amounts of ascites were observed.

  1. Is the operation done by total laparoscope? Including intra-coporeal end to end anastomosis? Is there any discrepancy between the proximal and distal bowel?

Response: As you pointed out, it was not laparoscopic surgery, but laparoscopic-assisted surgery. Surgeons identified the lesion of the ileum laparoscopically, and resection of the ileum including the Meckel’s diverticulum and end-to-end anastomosis were performed extra-corporeally. There was no discrepancy between the proximal and distal bowel. We have revised our text as follows:

Page 3, lines 109-110: the patient underwent laparoscopic-assisted surgery by pediatric surgeons

Page 3, line 115: the ileum was anastomosed end-to-end extra-corporeally.

Page 5, lines 158-160: Finally, laparoscopic-assisted resection of a pedunculated polyp-like structure identified at approximately 90 cm proximal to the ileocecal valve,

  1. Why did you choose to do the operation about two weeks later? Why didn't you do the operation urgently? Do you have to worry about the possibility of bowel ischemia or worsening of the patient's condition?

 Response: In fact, we had wanted to perform the surgical intervention as soon as possible. However, we decided to perform semi-emergent surgery instead of emergent surgery because the patient had a temporary small intestinal intussusception with spontaneous resolution of symptoms both on ultrasound and clinical findings, and the patient had presented with similar episodes. In addition, parental consent was not immediately obtained. We have thus revised the manuscript as follows, consistent with the suggestions of reviewer 2:

Page 3, line 105-108: It was decided to perform semi-emergent surgery rather than emergent surgery because the patient had a temporary small intestinal intussusception with spontaneous resolution of symptoms on ultrasonographic and clinical findings, with similar episodes, and parental consent could not be obtained immediately. Approximately two weeks after

  1. Because a pathologic leading point is highly suspected in this case, do you think double-balloon endoscopy is necessary? Is the exam will affect your further treatment plan? If the DBE is negative in this case, what will you do next? Is there any suggestion you want to give to the readers?

Response: An isolated, pedunculated, small intestinal polyp was also considered as a differential diagnosis in the present case. In the case of small intestinal polyps, small intestinal balloon endoscopy is useful for performing polypectomy without invasive surgery (Krasaelap A, et al. J Pediatr Gastroenterol Nutr. 2020;71(4):491-3). In addition, the presence of small intestinal polyps may suggest the possibility of juvenile polyposis syndrome (Krasaelap A, et al. J Pediatr Gastroenterol Nutr. 2020;71(4):491-3), and small intestinal balloon endoscopy may also contribute to screening for such a disorder. Endoscopic resection of an inverted Meckel's diverticulum may also be possible in the future with the establishment of small intestinal balloon endoscopy in pediatric populations. We have thus revised the text as follows:

Page 7, lines 220-226: However, the differential diagnosis of inverted Meckel's diverticulum with a mass-like leading point, as in the present case, includes isolated small intestinal pedunculated polyps. In the case of small intestinal polyps, small intestinal balloon endoscopy is useful in performing polypectomy without invasive surgery [38]. In addition, the presence of small intestinal polyps may suggest the possibility of juvenile polyposis syndrome [38], and small intestinal balloon endoscopy may also contribute to screening for such a disorder.

Page 7, lines 230-231: resulting in prevention of intestinal perforation and obstruction and exclusion of a small intestinal polyp and polyposis.

Page 9, lines 335-336: 38. Krasaelap, A.; Lerner, D.; Southern, J.; Noe, J.; Chugh, A. Endoscopic Removal of a Single, Painless, Juvenile Polyp in the Small Intestine Causing Anemia. J Pediatr Gastroenterol Nutr. 2020, 71, 491-493.

Reviewer 2 Report

The present paper aims to present a particular case of Meckel diverticulum complicated by inversion and intermitent intussusception. The case presentation is well documented and complete.

The associated review, although short, is concise and provides enough information for comparing similar cases in adult and pediatric patients. It covers some epidemiological data, clinical featurea, imaging diagnosis, management and pathological findings.

However, I have sme questions:

1. Why are you insisting in presenting hematemesis as a sign of intestinal bleeding, under the Treitz angle ?

2. What happened to the patirn during the 2 weeks before surgery ? Why did you decide to delay the surgical intervention ?

3. Table 1: It would be intresting to know how many diverticulectomies and segmental ilal resections were. The isolated diverticulectomy may expose to the risk of remaning gastric/pancreatic cells in the MD base.

4. The Abstract and the Conclusion chapters should be revised, as well as the English language.

Author Response

Reviewer 2

The present paper aims to present a particular case of Meckel diverticulum complicated by inversion and intermitent intussusception. The case presentation is well documented and complete.

The associated review, although short, is concise and provides enough information for comparing similar cases in adult and pediatric patients. It covers some epidemiological data, clinical featurea, imaging diagnosis, management, and pathological findings.

However, I have some questions.

Response: Thank you very much for your helpful comments on our paper. We have revised our manuscript based on the suggestions provided. The corrections are shown in red in this letter and our revised manuscript. We hope that you will find these revisions satisfactory.

Specific Comments:

  1. Why are you insisting in presenting hematemesis as a sign of intestinal bleeding, under the Treitz angle?

Response: As pointed out, "hematemesis" was used incorrectly, when in fact we meant "bloody stool". We have changed all the relevant parts as follows:

Page 1, lines 11-12: gastrointestinal symptoms other than bloody stool.

Page 1, line 32: symptoms such as acute abdominal pain, vomiting, and, especially, bloody stool.

Page 1, lines 36-37: symptoms other than bloody stool [1]. Therefore, there are only a few cases of children with abdominal pain and vomiting without bloody stool

Page 6, line 166: cases are diagnosed by bloody stool and obstructive bowel symptoms

Page 6, line 176: no obvious bloody stool was evident,

Page 6, line 178: bloody stool was observed in 55/74 (74.3%) of adults

Page 6, line 181: because of the absence of bloody stool, eight pediatric cases,

  1. What happened to the patient during the 2 weeks before surgery? Why did you decide to delay the surgical intervention?

Response: In fact, we had wanted to perform the surgical intervention as soon as possible. However, we decided to perform semi-emergent surgery instead of emergent surgery because the patient had a temporary small intestinal intussusception with spontaneous resolution of symptoms both on ultrasound and clinical findings, and the patient had presented with similar episodes. In addition, parental consent was not immediately obtained. We have thus revised the manuscript as follows, consistent with the suggestions of reviewer 1:

Page 3, line 105-108: It was decided to perform semi-emergent surgery rather than emergent surgery because the patient had a temporary small intestinal intussusception with spontaneous resolution of symptoms on ultrasonographic and clinical findings, with similar episodes, and parental consent could not be obtained immediately. Approximately two weeks after

  1. Table 1: It would be interesting to know how many diverticulectomies and segmental ileal resections were. The isolated diverticulectomy may expose to the risk of remaining gastric/pancreatic cells in the MD base.

Response: Unfortunately, in this review, 72 patients were reported to have undergone either diverticulectomies or segmental ileal resections, but in several cases, which procedure had been performed was not described; Table 1 shows the information available.

  1. The Abstract and the Conclusion chapters should be revised, as well as the English language.

Response: Based on the separately attached data file, we have revised the text as follows:

Page 1, lines 12-14: A nine-year-old boy diagnosed as having cyclic vomiting because of recurrent abdominal pain and vomiting for 6 months was referred to our institute and hospitalized due to worsening symptoms. After admission, abdominal ultrasonography showed

Page 7, lines 228-239: A case of inverted Meckel’s diverticulum that mimicked cyclic vomiting due to recurrent abdominal pain and vomiting was presented. This case was promptly diagnosed by DBE, resulting in prevention of intestinal perforation and obstruction and exclusion of a small intestinal polyp and polyposis. In cases involving periodic attacks of vomiting and abdominal pain, unnecessary emergent surgery could be avoided by cautious imaging evaluation, considering ileo-ileal intussusception with an advanced lesion of an inverted Meckel’s diverticulum as a differential diagnosis, without facilely diagnosing cyclic vomiting. In addition, the particularity of this case consists in the fact that the inverted Meckel’s diverticulum didn't determine an evolutive intussusception in a child patient. Thus, the necessity of an emergent surgery is not always related to a thorough clinical and imaging evaluation, but to the possibility of an inverted Meckel’s diverticulum to cause an evolutive intussusception, especially in adults.

  1. Lines 40-42: "a few reports have demonstrated the efficacy of double-balloon endoscopy (DBE) for hemorrhage of pediatric Meckel's diverticulum" - efficacy for the diagnosis or for the treatment?

Response: We have revised it to state that it was for diagnosis, as follows:

Page 1, lines 41-42: a few reports have demonstrated the efficacy of double-balloon endoscopy (DBE) for the diagnosis of hemorrhage in pediatric Meckel's diverticulum

  1. Lines 46-48: The sentence should be reviewed (e.g: the word "which" seems to be related to the word "ultrasnography", not to "a case of", which leads to confusion.

Response: We have revised it to make it clear, as follows:

Page 2, line 46-48: In this report, a case of cyclic vomiting-like symptoms such as periodic abdominal pain and vomiting with inverted Meckel's diverticulum diagnosed using transanal DBE after abdominal ultrasonography is presented.

  1. Line 68, 76: AT admission
  2. Line 117: It should be better to to say that the patient (not the case) was diagnosed as having ...
  3. Line 199: I suggest to replace "could be performed" with “had been performed”.

Response: The suggested revisions have been made.

  1. Lines 157-160: The sentence should be reviewed. The inverted MD was performed laparoscopically or by DBE? The definitive diagnosis of MD was established during the surgery, or anatomopathological?

Response: We have revised the text to make it clear, as follows:

Page 5, lines 158-161: Finally, laparoscopic-assisted resection of a pedunculated polyp-like structure identified at approximately 90 cm proximal to the ileocecal valve, which is a preferred site for Meckel’s diverticulum, led to the definitive diagnosis of inverted Meckel’s diverticulum with histological findings of ectopic gastric mucosa.

  1. Which affirmation remains in the article?

Lines 33-34: approximately 30% of Meckel's diverticulum cases show clinical symptoms throughout their life

Lines 166-167: Symptomatic Meckel's diverticulum cases are estimated to account for about four to six percent of all Meckel's diverticulum cases

Response: We have revised the text, as follows.

Page 6, lines 167-168: Symptomatic Meckel's diverticulum cases are estimated to account for about 30% of all Meckel's diverticulum cases

  1. Line 172: the words "atypical disease age of onset", lead to misunderstanding.

Response: We have revised the text, as follows:

Page 6, line 173: cases with an atypical onset age and the only obvious clinical

  1. Lines 207-210: Ultrasonography is a useful tool in the diagnosis of intussusception, especially in children. But is the abdominal ultrasound able to diagnose the cause of the intussusception, especially an inverted MD? Maybe the sentence should be reviewed.

Response: As pointed out, ultrasonography is only useful in the diagnosis of intussusception and some advanced pathological lesions. We have revised our text, as follows:

Page 6, lines 208-210: abdominal ultrasonography is not only radiation-free and noninvasive compared to abdominal CT, but it is also a useful tool in the diagnosis of intussusception caused by advanced pathological lesions, including inverted Meckel's diverticulum in children. 

Reviewer 3 Report

Case report must be improved, it must be more accurate, for example, presenting data from laboratory with the normal value, summarizing the patient's history only with relevant information.

The conclusion must be improved, you made a good research but you even mentioned it in the conclusion, you should recapitulate your data and try to give the reader some guidelines for this type of patient/case.

Author Response

Reviewer 3

Response: Thank you very much for your helpful comments on our paper. We have revised our manuscript based on the suggestions provided. The corrections are shown in red in this letter and our revised manuscript. We hope that you will find these revisions satisfactory.

Case report must be improved, it must be more accurate, for example, presenting data from laboratory with the normal value, summarizing the patient's history only with relevant information.

Response: As suggested, we have added the reference ranges at our institute for the laboratory values in the “Case Report” section. We have also revised the text based on the points suggested by the other reviewers.

Page 3, lines 56-64: The patient was admitted to our institute with the chief complaint of a six-month history of recurrent abdominal pain and vomiting. He had been diagnosed as having cyclic vomiting by his family physician and referred to our institute for further treatment. At the time of referral, the patient was in the intermittent phase, and the fecal hemoglobin (Hb) level was 29 [reference range (RR) 0-100] ng/mL. Therefore, considering the possibility that the cyclic vomiting may resolve during the natural course of the disorder, the patient was followed without treatment. However, his symptoms worsened two days before admission, and he was admitted to our institute on the same day he visited as an outpatient.

Page 2, lines 73-76: Laboratory findings at admission were as follows: hemoglobin 13.9 (RR 12.6-16.5) g/dL and ferritin of 46 (RR 25-280) ng/mL, indicating no iron deficiency anemia; total protein 6.5 (RR 6.3-7.8) g/dL, and albumin 4.4 (RR 3.8-4.8) g/dL within normal ranges; and inflammatory markers not increased.

The conclusion must be improved, you made a good research but you even mentioned it in the conclusion, you should recapitulate your data and try to give the reader some guidelines for this type of patient/case.

Response: We have revised the “Conclusion” according to the suggestions of the reviewers as follows

Page 7, lines 228-239: A case of inverted Meckel’s diverticulum that mimicked cyclic vomiting due to recurrent abdominal pain and vomiting was presented. This case was promptly diagnosed by DBE, resulting in prevention of intestinal perforation and obstruction and exclusion of a small intestinal polyp and polyposis. In cases involving periodic attacks of vomiting and abdominal pain, unnecessary emergent surgery could be avoided by cautious imaging evaluation, considering ileo-ileal intussusception with an advanced lesion of an inverted Meckel’s diverticulum as a differential diagnosis, without facilely diagnosing cyclic vomiting. In addition, the novelty of this case consists in the fact that the inverted Meckel’s diverticulum did not result in an evolving intussusception in a child patient. Thus, the necessity of emergent surgery is not always related to a thorough clinical and imaging evaluation, but to the possibility of an inverted Meckel’s diverticulum causing an evolving intussusception, especially in adults.
